# CrossFusion: A Multi-Scale Cross-Attention Convolutional Fusion Model for Cancer Survival Prediction

**Rustin Soraki**[1]                                    RUSTIN@CS.WASHINGTON.EDU

**Huayu Wang**[1]                                        HUAYU@UW.EDU

**Sitong Liu**[1]                                        SITONL2@UW.EDU

**Joann G. Elmore**[2]                          JELMORE@MEDNET.UCLA.EDU

**Linda Shapiro**[1]                          SHAPIRO@CS.WASHINGTON.EDU

[1] *University of Washington, Seattle, WA*

[2] *University of California, Los Angeles, CA*

**Editors:** Accepted for publication at MIDL 2026

## Abstract

Cancer survival prediction from whole slide images (WSIs) relies on capturing prognostic features spanning multiple magnifications, from global tissue architecture to fine-grained cellular morphology. However, current approaches typically face two main limitations: most frameworks focus heavily on single-scale analysis, thereby overlooking the hierarchical context of tissue; meanwhile, existing multi-scale methods often employ simplistic fusion mechanisms (e.g., direct concatenation) that fail to model effective cross-scale interactions. To address these challenges, we propose CrossFusion, a novel multi-scale architecture that introduces a convolutional fusion processor to perform rigorous scale–space integration. Evaluated on six TCGA cancer cohorts, CrossFusion achieves state-of-the-art C-index performance, consistently outperforming both strong single-scale and multi-scale baselines. Furthermore, leveraging domain-specific pathology feature extractors yields additional gains in prognostic accuracy compared to general-purpose backbones. The source code is available at: https://github.com/RustinS/CrossFusion

**Keywords:** Computer Vision, Computational Pathology, Survival Prediction, Cross-Attention, Multi-Scale Image Processing

## 1. Introduction

Whole Slide Images (WSIs) capture critical tumor characteristics and are central to modern cancer diagnosis (Kumar et al., 2020; Kothari et al., 2013; Ghaznavi et al., 2013; Liu et al., 2025). In clinical practice, survival analysis based on WSIs plays a vital role in informing prognosis and guiding treatment strategies (Campanella et al., 2019). Recent advances in survival analysis have leveraged multiple-instance learning (MIL) and deep learning. For instance, Ilse et al. (Ilse et al., 2018) employ attention mechanisms to identify the most predictive regions, while Li et al. (Li et al., 2021) aggregate instance-level features into robust slide-level representations. Transformer-based approaches (Shao et al., 2021) capture long-range dependencies and graph-based methods (Li et al., 2018; Chen et al., 2021) model spatial relationships effectively. More recently, methods by Yang et al. (Yang et al., 2024) and Wu et al. (Wu et al., 2024) have enhanced interpretability and prediction by integrating sparse attention and prototypical representations. Despite these advancements,

the enormous size and heterogeneity of WSIs make it challenging for AI models to capture high-level global tissue patterns and fine-grained cellular detail, both essential for robust survival analysis.

A multi-scale approach offers a promising solution by combining large patches that provide an overview of tissue architecture with small patches that capture detailed cellular morphology. By merging coarse structural cues with fine-grained details, multi-scale methods can better reflect the complex biological processes underlying tumor development and progression. Prior studies have shown that integrating information across multiple scales significantly improves diagnostic accuracy and reduces errors (Tan et al., 2023). For example, Deng et al. (Deng et al., 2024) employ cross-scale attention maps to aggregate features, while Wu et al. (Wu et al., 2021) use features from different scales as keys, queries, and values to guide learning. Similarly, Zhao et al. (Zhao et al., 2024) select informative patches using a variational positive-unlabeled framework and fuse them with cross-attention. However, these approaches often overlook certain resolution levels or rely on suboptimal fusion techniques, leaving two key challenges unresolved: effectively combining complementary information from multiple scales and developing robust methods for fusing these features.

To address these challenges, we propose **CrossFusion**, a novel framework that unifies multi-scale patch embeddings from WSIs into a single, predictive representation. We summarize our main contributions as follows:

1. **Multi-Scale Cross-Attention:** This module enables interaction between features at different resolutions, allowing high-resolution details and low-resolution global patterns to reinforce each other while preserving spatial context.

2. **Dual-Path Global–Local Context Alignment:** Rather than treating multi-scale fusion as a purely attention-driven or convolution-driven problem, we propose a dual-path global–local alignment mechanism, where transformers capture cross-scale global dependencies convolutions enforce spatial alignment and local coherence.

3. **Extensive Validation & Accuracy:** We validate CrossFusion on diverse cancer survival datasets, demonstrating that it matches or exceeds state-of-the-art performance in survival analysis while maintaining interpretability through visualization of key regions at multiple magnifications. We also examine the effect of different feature extraction backbones and compare CrossFusion trained on the domain-specific backbones to the general one.

## 2. Related Work

To process these gigapixel-resolution images, Multiple Instance Learning (MIL) has become the standard paradigm. In this framework, a WSI is treated as a "bag" of smaller patches (instances), and a slide-level prediction is aggregated from patch-level features. While early methods focused on simple aggregation, recent advancements have introduced attention mechanisms, graph convolutional networks (GCNs), and transformers to better capture the spatial and semantic dependencies between tissue patches.

## 2.1. Single-Scale Computational Pathology Methods

Most state-of-the-art methods operate on a single magnification scale. Attention-based models like AMIL (Ilse et al., 2018) and DSMIL (Li et al., 2021) aggregate patch features by identifying predictive regions. Graph-based approaches, such as DeepGraphSurv (Li et al., 2018) and Patch-GCN (Chen et al., 2021), model spatial topology , while Transformer-based methods like TransMIL (Shao et al., 2021) capture long-range dependencies. We also compare against recent specialized architectures like SCMIL (Yang et al., 2024) and ProtoSurv (Wu et al., 2024), which utilize sparse attention and prototypical representations to enhance interpretability and performance.

## 2.2. Multi-Scale Histopathology Methods

Multi-scale methods aim to mimic the pathologist's workflow by integrating coarse structural cues with fine-grained cellular details. Existing frameworks have adopted distinct strategies to achieve this: ZoomMIL (Thandiackal et al., 2022) selectively mines salient regions for high-magnification feature extraction, whereas CSMIL (Deng et al., 2024) employs convolutional networks to fuse multi-scale features. Additionally, MuSTMIL (Marini et al., 2021) leverages multi-task learning to maximize data utilization, and HIPT (Chen et al., 2022) utilizes a feature pyramid approach to derive hierarchical representations for survival tasks.

However, while prior works have employed mechanisms such as cross-scale attention maps or scale-specific key-query interactions, they often overlook explicit connections between resolutions or rely on suboptimal fusion techniques. CrossFusion addresses these limitations by using multi-magnification patch features to explicitly model "intra-scale and inter-scale" interactions. Through a novel integration of cross-scale cross-attention, convolutional fusion, and Transformer encoding, our framework directly outputs time-series survival risks, demonstrating significant performance gains over existing methods across multiple cancer cohorts.

## 3. Method

This section outlines the pipeline of our proposed methodology, CrossFusion. Figure 1 illustrates the complete framework, including its main stages and components. As shown in the figure, the extracted patches at different magnifications are encoded by a feature extractor, projected into a common embedding space, and then through the Cross-Attention Block, in which the different magnifications interact. The next step is the Pad-Transformer process, which uses Pyramid Position Encoding to capture local and global context. The Conv Processor is used to fuse the multi-scale features.

### 3.1. CrossFusion

The CrossFusion module takes three inputs: $\mathbf{X}_C$ (coarse), $\mathbf{X}_S$ (source), and $\mathbf{X}_F$ (fine), which represent patch embeddings from 5x, 10x, and 20x magnifications, respectively. Initially, each embedding is projected into a shared space $D_e$. Next, to facilitate inter-scale interactions, the module applies cross-attention with $\mathbf{X}_S$ as the query and the other embeddings as context:

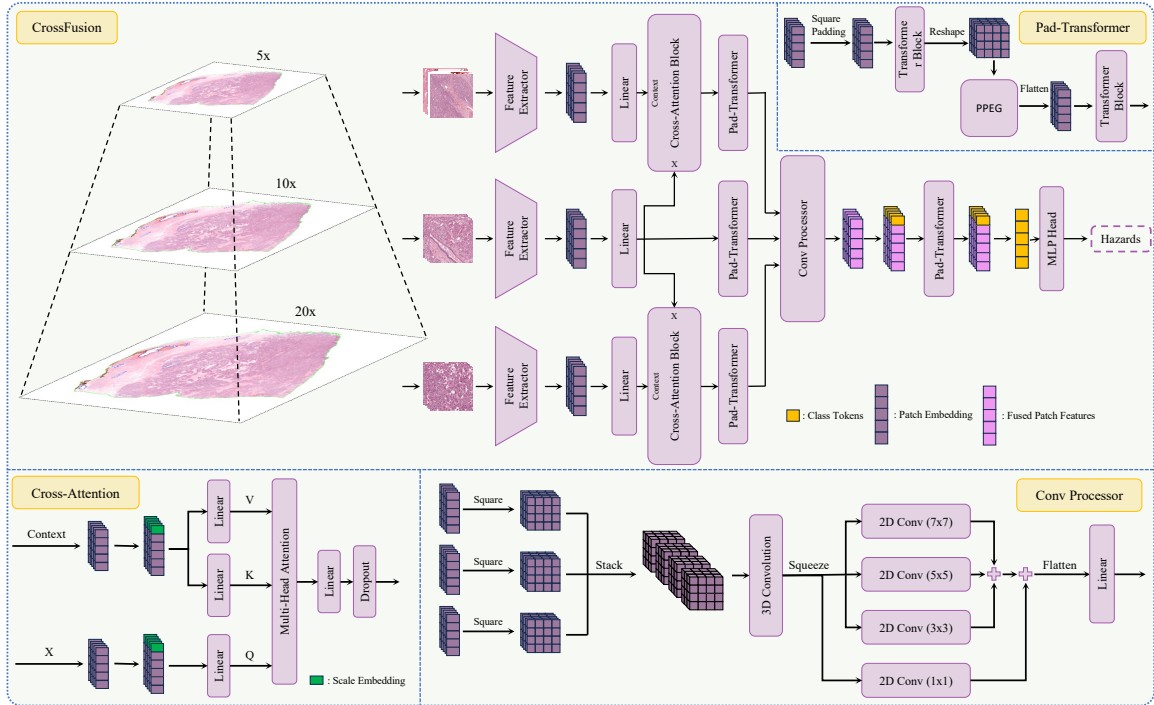

Figure 1: Overview of CrossFusion. WSIs are processed by extracting patches at 5x (coarse), 10x (source), and 20x (fine) magnifications, which are first encoded using a feature extractor and then projected into a common embedding space. The source features interact with the coarse and fine features via cross-attention blocks, and each branch is refined by Pad-Transformers. The multi-scale features are subsequently fused using a Conv Processor, and a replicated learnable class token is appended. An additional transformer block refines this token, and an MLP head produces the final survival predictions from the class tokens.

$$\mathbf{X}'_{\mathrm{C}} = CAB(\mathbf{X}_{\mathrm{S}}, \mathbf{X}_{\mathrm{C}}), \quad \mathbf{X}'_{\mathrm{F}} = CAB(\mathbf{X}_{\mathrm{S}}, \mathbf{X}_{\mathrm{F}}) \tag{1}$$

where $CAB$ denotes the Cross-Attention Block. Each feature set is then processed by dedicated Pad-Transformer ($PT$) blocks. The outputs are fused via the Conv Processor ($CP$):

$$\mathbf{X}_{\mathrm{fused}} = CP\Big(PT(\mathbf{X}'_{\mathrm{C}}),\, PT(\mathbf{X}_{\mathrm{S}}),\, PT(\mathbf{X}'_{\mathrm{F}})\Big) \tag{2}$$

A learnable class token $\mathbf{c} \in \mathbb{R}^{1 \times D_e}$ is first replicated and prepended to the fused token sequence. This extended sequence is processed by an additional Pad-Transformer followed by Layer Normalization. The class token is then extracted to yield $\mathbf{c}'$. An MLP head maps $\mathbf{c}'$ to logits $\mathbf{l}$, from which hazards $\mathbf{h}$ are computed using a sigmoid activation. Finally, survival probabilities $\mathbf{S}$ are obtained as the cumulative product of $1 - \mathbf{h}$.

### 3.2. Cross-Attention Block

The cross-attention module fuses information from two inputs: the primary input $\mathbf{X} \in \mathbb{R}^{B \times N \times D}$ and a contextual input $\mathbf{Context} \in \mathbb{R}^{B \times M \times D}$, where $B$ is the batch size, $N$ and $M$ are sequence lengths, and $D$ is the embedding dimension. Both inputs are augmented with a learnable scale embedding $\mathbf{s} \in \mathbb{R}^{1 \times 1 \times D}$ before computing attention, acting like a learnable positional encoding. The queries, keys, and values are obtained via linear projections:

$$\mathbf{Q} = W_q(\mathbf{X} + \mathbf{s}), \quad \mathbf{K} = W_k(\mathbf{Context} + \mathbf{s}), \quad \mathbf{V} = W_v(\mathbf{Context} + \mathbf{s}), \tag{3}$$

where $W_q, W_k, W_v \in \mathbb{R}^{D \times D}$ are learnable weight matrices. Multi-head attention is then applied, followed by an output projection. The output is further processed through residual connections, layer normalization, and a feed-forward network. This mechanism enables effective information exchange between the primary input and its contextual counterpart, improving feature representation.

### 3.3. Pad-Transformer

The Pad-Transformer organizes input tokens into a grid, processes them with an initial Transformer block, uses Pyramid Position Encoding Generator (PPEG) (Shao et al., 2021) to add spatial context, and then further refines the features using a second Transformer block. This combines global dependencies (from the first block) and local details (via PPEG) before final refinement by the second Transformer block.

#### 3.3.1. SQUARE PADDING

Given an input sequence $\mathbf{X} \in \mathbb{R}^{B \times N \times D}$, where $N$ is the number of tokens, we first compute $H = W = \lceil \sqrt{N} \rceil$. We then pad $\mathbf{X}$ by appending the first $(H \times W) - N$ tokens to the end of the sequence, resulting in a sequence of length $H \times W$. Finally, this padded sequence is reshaped into a square grid $\mathbf{X}_s \in \mathbb{R}^{B \times H \times W \times D}$ for subsequent spatial operations.

#### 3.3.2. TRANSFORMER BLOCKS

Each transformer block applies layer normalization, computes multi-head self-attention, and adds the result to the input via a residual connection. A second layer normalization is followed by a feed-forward network, GELU activation, and dropout.

#### 3.3.3. CONVOLUTIONAL POSITIONAL ENCODING (PPEG)

To incorporate local spatial context, the PPEG module applies three parallel depth-wise convolutions with kernel sizes 7, 5, and 3 to the reshaped feature map $\mathbf{X}_s \in \mathbb{R}^{B \times D \times H \times W}$:

$$\mathbf{X}_{\text{PPEG}} = \text{Conv}_7(\mathbf{X}_s) + \text{Conv}_5(\mathbf{X}_s) + \text{Conv}_3(\mathbf{X}_s) + \mathbf{X}_s \tag{4}$$

This operation enhances token representations with detailed local positional information.

### 3.4. Conv Processor

The Conv Processor fuses multi-source features and enhances spatial representations using multi-scale convolutions. The combination of Transformer and CNN leverages global and local modeling strengths: Transformer captures long-range dependencies while CNN reinforces local spatial features afterward, enabling multi-source information fusion and compensating for each other's limitations. Given three input sequences $\mathbf{X}_i \in \mathbb{R}^{B \times N \times D}, \quad i \in \{1, 2, 3\}$, each is square-padded and reshaped into a 2D feature map, $\mathbf{X}'_i \in \mathbb{R}^{B \times D \times H \times W}$, where $N = H \times W$. The feature maps are stacked into $\mathbf{X}_{\text{stack}} \in \mathbb{R}^{B \times 3 \times D \times H \times W}$ and fused via a 3D convolution to get $\mathbf{X}_{\text{fused}}$. After squeezing the singleton channel, multi-scale features are extracted by applying parallel depth-wise convolutions with kernel sizes 7, 5, 3, and 1, where the output dimension is reduced to $D' = D//2$:

$$\mathbf{X}_{\text{ms}} = \text{Conv}_7(\mathbf{X}_{\text{fused}}) + \text{Conv}_5(\mathbf{X}_{\text{fused}}) + \text{Conv}_3(\mathbf{X}_{\text{fused}}) + \text{Conv}_1(\mathbf{X}_{\text{fused}}). \tag{5}$$

The resulting feature map is flattened along spatial dimensions into $\mathbf{X}_{\text{flat}} \in \mathbb{R}^{B \times D' \times (H \cdot W)}$ and permuted into a token sequence $\mathbf{X}_{\text{seq}} \in \mathbb{R}^{B \times (H \cdot W) \times D'}$. Finally, a linear projection followed by Layer Normalization restores the original dimension $(D)$. This module efficiently fuses multi-source information while capturing multi-scale spatial features.

## 4. Experimental Setup

### 4.1. Dataset

We used H&E WSIs from six TCGA cancer types: BLCA (437 slides), BRCA (1016 slides), COAD (424 slides), GB&LGG (1041 slides), LUAD (507 slides), and UCEC (539 slides). These datasets were chosen for their size, public availability, survival follow-up data, and a balanced uncensored-to-censored ratio (average 0.28). On average, each WSI yields 13,496 patches at 20x, 3,449 patches at 10x, and 895 patches at 5x, with the number of 20x patches reaching up to 137,990.

### 4.2. Implementation Details

**Patch Extraction and Embedding:** We used CLAM (Lu et al., 2021) to extract 256×256 patches at 20x, 10x, and 5x magnifications and extract features from different feature extraction backbones. Tissue regions were identified using a binary mask computed by thresholding the saturation channel in HSV.

**Training and Evaluation:** The model was trained with Adam (learning rate $1 \times 10^{-4}$, weight decay $4 \times 10^{-6}$, batch size 1) with a 5-epoch warm-up, and evaluated via 5-fold cross-validation. For a fair comparison, all methods used the same loss function, feature embeddings, and hyperparameters. Experiments were implemented in PyTorch on a workstation with four Nvidia RTX A4000 GPUs.

**Evaluation Metrics:** Performance was measured using the mean C-index across validation splits. Additionally, we report the p-value from stratifying patients into high- and low-risk groups as a statistical measure of the model's discriminative ability.

Table 1: C-Index (mean $_{std}$) of different methods over the six different datasets. The best and the second-best results are highlighted in **bold** and underline, respectively.

| | BLCA | BRCA | COAD | GB&LGG | LUAD | UCEC |
|---|---|---|---|---|---|---|
| *Single Scale Methods* | | | | | | |
| AMIL | $.559_{059}$ | $.590_{050}$ | $.662_{063}$ | $.759_{111}$ | $.590_{036}$ | $.644_{092}$ |
| DSMIL | $.552_{050}$ | $.564_{044}$ | $.610_{012}$ | $.728_{102}$ | $.579_{032}$ | $.601_{073}$ |
| TransMIL | $.574_{064}$ | $.594_{045}$ | $.656_{057}$ | $.772_{093}$ | $.594_{059}$ | $.664_{044}$ |
| DeepGraphSurv | $.572_{054}$ | $.558_{099}$ | $.591_{119}$ | $.764_{053}$ | $.622_{055}$ | $.635_{061}$ |
| PatchGCN | $.563_{043}$ | $.595_{089}$ | $.612_{144}$ | $.774_{046}$ | $.577_{081}$ | $.679_{071}$ |
| SCMIL | $.566_{054}$ | $.590_{034}$ | $\underline{.677}_{070}$ | $.763_{094}$ | $.584_{050}$ | $.668_{071}$ |
| ProtoSurv | $.579_{023}$ | $.627_{034}$ | $.668_{057}$ | $.776_{031}$ | $.619_{046}$ | $\mathbf{.730_{032}}$ |
| *Multi Scale Methods* | | | | | | |
| ZoomMIL | $.570_{056}$ | $.563_{047}$ | $.642_{066}$ | $.770_{091}$ | $.568_{046}$ | $.679_{033}$ |
| MuSTMIL | $.575_{065}$ | $.589_{043}$ | $.640_{084}$ | $.780_{089}$ | $.600_{040}$ | $.682_{039}$ |
| CSMIL | $.542_{071}$ | $.589_{070}$ | $.636_{087}$ | $.742_{119}$ | $.582_{060}$ | $.640_{047}$ |
| HIPT | $.554_{015}$ | $.603_{045}$ | $.651_{102}$ | $.773_{032}$ | $.572_{035}$ | $.697_{060}$ |
| *Ours* | | | | | | |
| CrossFusion w/o CP | $\underline{.627}_{014}$ | $\underline{.631}_{076}$ | $.669_{069}$ | $\underline{.787}_{081}$ | $\underline{.627}_{038}$ | $\underline{.710}_{061}$ |
| CrossFusion w/o F&C | $.562_{058}$ | $.629_{052}$ | $.631_{052}$ | $.782_{074}$ | $.609_{053}$ | $.672_{050}$ |
| **CrossFusion** | $\mathbf{.630_{027}}$ | $\mathbf{.643_{037}}$ | $\mathbf{.694_{053}}$ | $\mathbf{.797_{056}}$ | $\mathbf{.627_{040}}$ | $.702_{044}$ |

## 5. Experiments and Results

In this section, we evaluate our model's performance through experiments. First, we compare CrossFusion to state-of-the-art methods. Next, we assess our model's interpretability by analyzing attention-based heatmaps, providing insights into its decision-making process. Finally, we analyze the effect of using different foundational models as feature extraction backbones to determine whether domain-specialized backbones improve performance.

### 5.1. Comparison with State-Of-The-Art Methods

We evaluated CrossFusion against state-of-the-art survival prediction methods, categorized into single-scale and multi-scale approaches. For single-scale methods, we compared against AMIL (Ilse et al., 2018), DSMIL (Li et al., 2021), TransMIL (Shao et al., 2021), Deep-GraphSurv (Li et al., 2018), Patch-GCN (Chen et al., 2021), SCMIL (Yang et al., 2024), and ProtoSurv (Wu et al., 2024). For multi-scale methods, which aim to mimic the pathologist's workflow by integrating coarse structural cues with fine-grained cellular details, we compared against ZoomMIL (Thandiackal et al., 2022), MUSTMIL (Marini et al., 2021), HIPT (Chen et al., 2022)and CSMIL (Deng et al., 2024). All models used ResNet50 (He et al., 2016) as the feature extractor for fair comparison.

As shown in Table 1, CrossFusion achieves the best or near-optimal performance across six cancer datasets. In the UCEC dataset, the low uncensored-to-all-slides ratio (0.15) posed a challenge due to CrossFusion's reliance on patch-level features without prior information, resulting in slightly lower performance than ProtoSurv, which leverages priors. Nevertheless, CrossFusion consistently outperforms all other baselines and remains compet-

itive with ProtoSurv, demonstrating robustness even under data constraints. Comparing with other multi-scale model, the superior performance suggests that CrossFusion functions not merely as a feature aggregator but as a scale-interrogative learner, enforcing consistency checks between tissue organization and cellular morphology under the intermediate 10x view.

Ablation studies validated the contributions of key components. First, replacing the ConvProcessor (CP) with simple concatenation/projection reduced performance on all datasets except UCEC (due to data scarcity), validating the effectiveness of our Global–Local Context Alignment module. Second, removing Fine (F) and Coarse (C) sources to rely solely on $20\times$ patches—following mainstream single-scale approaches like MMP (Song et al., 2024) and UniPro (Xu et al., 2025)—significantly degraded performance in most datasets, confirming the necessity of multi-scale inputs for capturing discriminative WSI features.

Finally, stratification analysis yielded p-values of $1.79 \times 10^{-4}$ for BLCA, $1.49 \times 10^{-2}$ for BRCA, $3.30 \times 10^{-4}$ for COAD, $2.30 \times 10^{-39}$ for GB&LGG, $1.92 \times 10^{-2}$ for LUAD, and $3.91 \times 10^{-3}$ for UCEC. These statistically significant results confirm that CrossFusion effectively differentiates high-risk and low-risk patient groups, underscoring its clinical relevance for survival prediction.

We further assessed CrossFusion's prognostic stratification by splitting patients into predicted high and low risk groups and visualizing their survival trajectories using Kaplan–Meier curves. As shown in Figure 2, the low-risk group (blue) consistently exhibits higher survival probabilities over time than the high-risk group (red) across all six TCGA cohorts (BLCA, BRCA, COAD, GM&LGG, LUAD, UCEC). The separation between the two curves is statistically significant in each cohort based on the log-rank test (all $p$-values $< 0.05$), with particularly strong discrimination observed in GM&LGG and additional clear separation in BLCA and COAD. These results indicate that CrossFusion learns risk scores that meaningfully stratify patients into subgroups with distinct survival outcomes.

### 5.2. Interpretability

To explore the model's decision-making, we generated heatmaps from attention weights in the Cross-Attention layers, the source features Pad-Transformer, and the fused features Pad-Transformer. Figure 3 shows a WSI from the TCGA-BRCA dataset—depicting a high-risk patient with low survival time—alongside its corresponding heatmaps.

The three intermediate heatmaps reveal that different modules capture distinct features: the Coarse Cross-Attention layer focuses on large-scale tissue organization, the Fine Cross-Attention layer captures detailed cellular morphology, and the Source Pad-Transformer emphasizes intermediate-scale structures. The final heatmap from the last transformer layer demonstrates that the model effectively filters out less relevant regions, concentrating on key histopathological features.

### 5.3. Analyzing the effect of Different Feature Extraction Backbones

We evaluate CrossFusion using different feature extraction backbones, comparing their impact on model performance. Specifically, we extract patch-level features using Conch (Lu et al., 2024), Uni2-h (Chen et al., 2024), QuiltNet (Ikezogwo et al., 2023), and Prov-GigaPath (Xu et al., 2024), and compare them against features extracted using ResNet50.

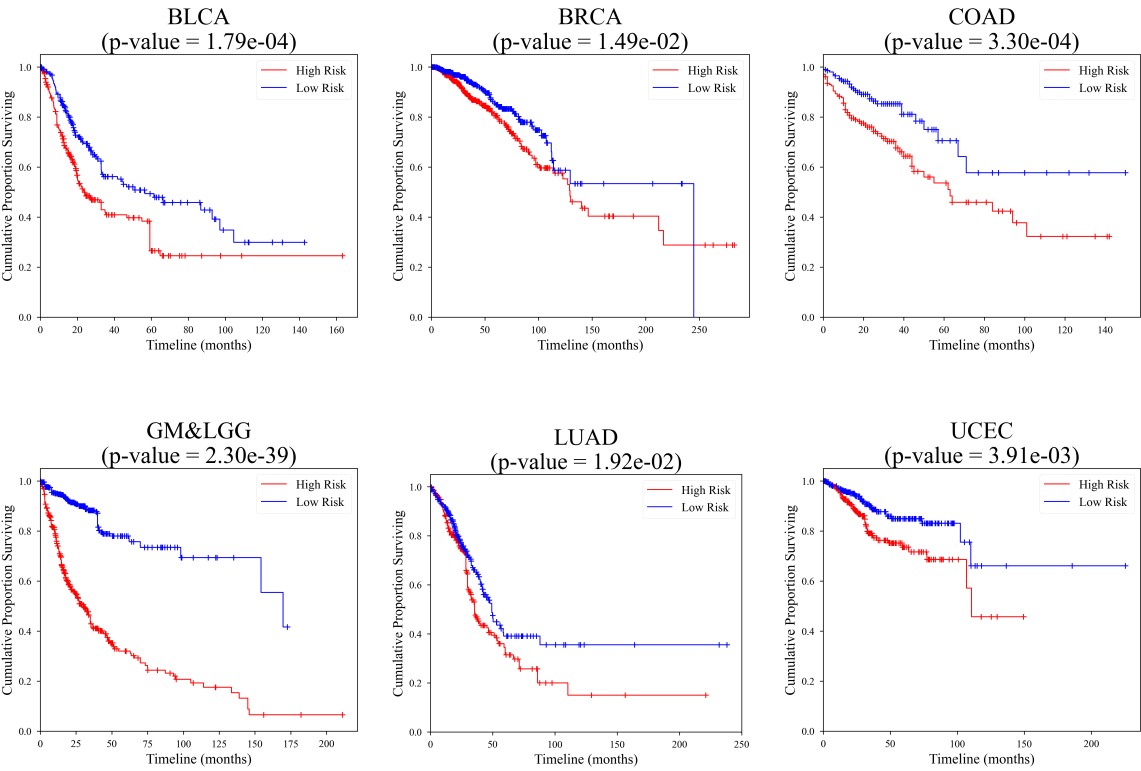

Figure 2: **Kaplan–Meier survival curves.** Kaplan–Meier curves of predicted high-risk (red) and low-risk (blue) groups across six TCGA cohorts. The log-rank test p-value is reported in each panel; $p < 0.05$ indicates statistically significant separation.

Table 2 shows that CrossFusion performs best with features from the Uni2-h backbone, while other domain-specific backbones yield similar results. The performance gap is highlighted particularly in the BRCA and the UCEC datasets, where utilizing high-quality features is crucial because of the low uncensored-to- all-slides ratio. The clear performance gap between CrossFusion trained on specialized backbones and CrossFusion trained on ResNet50 backbone highlights the benefit of domain-specific extraction backbones, which better capture tissue-level details, such as cellular morphology and tissue architecture, crucial for accurate prognostication.

While the transition to the Uni2-h backbone provides a general performance uplift across all methods, CrossFusion demonstrates the distinct ability to maximize the potential of these advanced representations. As shown in Table 3, our method consistently outperforms competing approaches under the same experimental settings. Specifically, CrossFusion not only surpasses the ResNet-based HIPT baseline by a substantial margin of 5.7% but, more importantly, maintains a clear lead over other multi-scale methods equipped with the same Uni2-h backbone (e.g., surpassing MuSTMIL by an average of 2.6%). This consistent superiority—particularly the 8.1% and 7.4% gains on BRCA and BLCA against HIPT—confirms

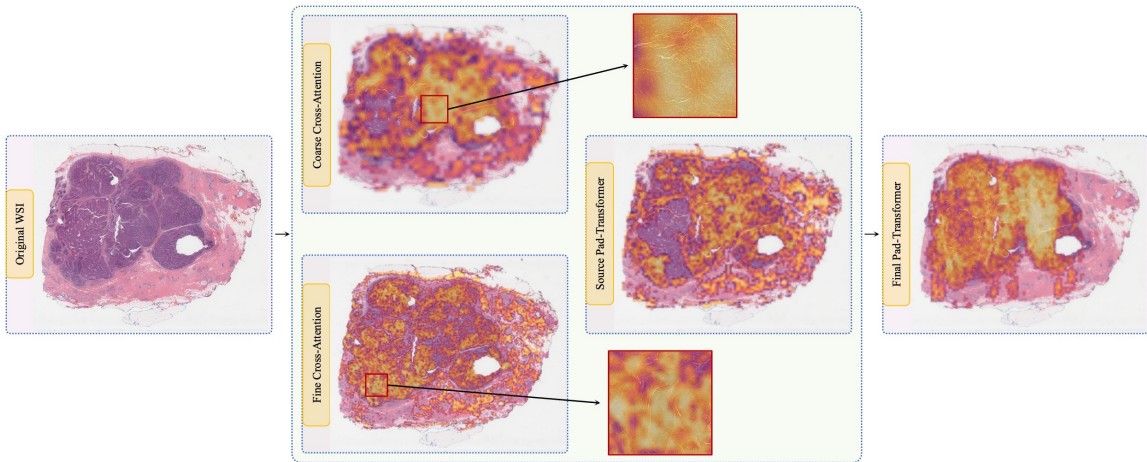

Figure 3: Generated heatmaps from the model predicting a high-risk case. The dark purple clusters mark tumor regions in the original WSI on the left and lighter yellow areas highlight important regions from the model's attention weights.

that the performance leap stems from our effective cross-scale fusion mechanism, rather than solely from the upgrade in feature extractors.

Table 2: C-Index (mean $_{std}$) of **CrossFusion** trained on different feature extraction backbones over the six different datasets. The best and the second-best results are highlighted in **bold** and underline, respectively.

|  | BLCA | BRCA | COAD | GB&LGG | LUAD | UCEC | Mean |
|---|---|---|---|---|---|---|---|
| w/ ResNet50 (Base) | .630$_{.027}$ | .643$_{.037}$ | .694$_{.053}$ | .797$_{.056}$ | .627$_{.040}$ | .702$_{.044}$ | .682 |
| w/ Conch | **.649$_{.053}$** | .675$_{.060}$ | .705$_{.024}$ | .799$_{.055}$ | .604$_{.055}$ | .737$_{.043}$ | .695 |
| w/ Uni2-h | .628$_{.019}$ | .684$_{.043}$ | .698$_{.023}$ | .810$_{.067}$ | .625$_{.051}$ | **.745$_{.030}$** | **.698** |
| w/ QuiltNet | .614$_{.050}$ | .650$_{.017}$ | .712$_{.043}$ | **.812$_{.032}$** | **.640$_{.064}$** | .727$_{.028}$ | .693 |
| w/ Prov-GigaPath | .635$_{.023}$ | **.686$_{.037}$** | **.718$_{.064}$** | .802$_{.051}$ | .620$_{.063}$ | .724$_{.043}$ | .698 |

## 5.4. Analysis of Inference Time

Inference Efficiency. We used the same 20x features in all methods. Our common feature extraction time was 30.61 seconds on average for 20x features in a Nvidia A4000 GPU. For 10x and 5x features, the CLAM can extract features in parallel, which will take less than 10 seconds depending on the number of foreground objects in the image. So we compare the inference time here. CrossFusion demonstrates superior efficiency with a slide-level fusion time of just 0.145 seconds. By eliminating the heavy computational bottlenecks found in graph-based methods (e.g., Patch-GCN: 40 seconds for graph construction) and hierarchical models (e.g., HIPT: 55 seconds), our approach is significantly faster. This

Table 3: Performance comparison (C-Index mean $_{std}$) of **CrossFusion** against other multi-scale MIL methods using **Uni2-h** features across six datasets.

|  | BLCA | BRCA | COAD | GB&LGG | LUAD | UCEC |
|---|---|---|---|---|---|---|
| MuSTMIL | .624$_{.047}$ | .634$_{.037}$ | .682$_{.086}$ | .793$_{.088}$ | .595$_{.056}$ | .708$_{.035}$ |
| CSMIL | .612$_{.036}$ | .619$_{.048}$ | .677$_{.065}$ | .806$_{.067}$ | .592$_{.037}$ | .716$_{.035}$ |
| **CrossFusion** | .628$_{.019}$ | .684$_{.043}$ | .698$_{.023}$ | .810$_{.067}$ | .625$_{.051}$ | .745$_{.030}$ |

low-latency inference makes CrossFusion highly suitable for large-scale, real-time clinical deployment.

## 6. Conclusion

We introduced CrossFusion, a novel framework that fuses multi-scale patch embeddings from WSIs using cross-attention, transformer-based spatial encoding, and convolutional fusion. By integrating multi-scale features, CrossFusion captures key histopathological patterns linked to patient survival. Our experiments on diverse TCGA cancer datasets show that CrossFusion demonstrates significant improvements over the current state-of-the-art survival prediction methods, even under challenging conditions.

Our results underscore the value of domain-specific feature extraction in preserving crucial tissue details, such as cellular morphology and tissue architecture. The attention-based heatmaps further confirm the model's effectiveness and offer insights into its decision-making process.

In summary, CrossFusion bridges advanced deep learning with clinical needs, providing a robust and interpretable tool for cancer survival prediction. Future work will explore additional data modalities to guide the model to focus on important case-specific patterns and enhance interpretability, paving the way for more personalized cancer treatment and improved patient outcomes.

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

## Appendix A. Failure Case Analysis

To provide a comprehensive evaluation of CrossFusion, we analyzed cases where the model failed to correctly stratify patient risk. Figure S1 and Figure S2 visualizes the attention heatmaps for two such representative failure cases.

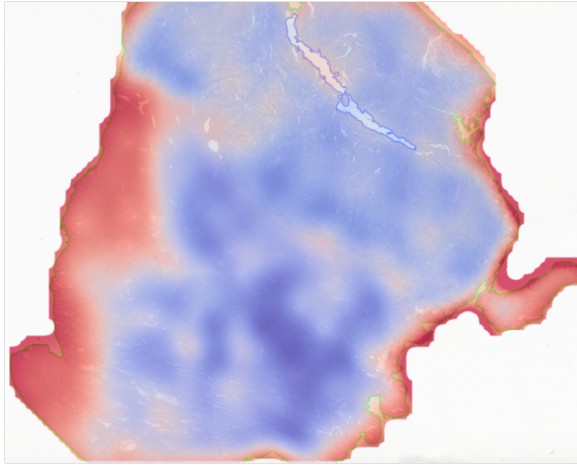

Figure S1: Failure Case - Low Risk.

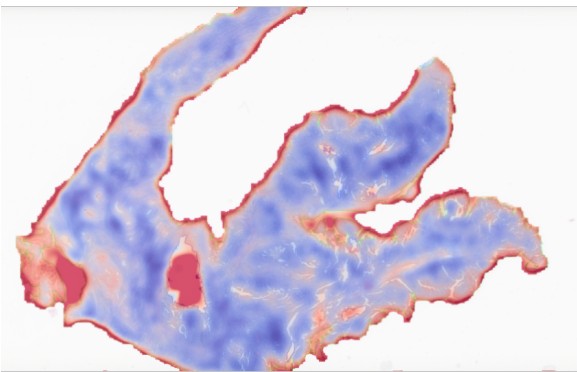

Figure S2: Failure Case - High Risk.

**Observation:** As observed in the heatmaps, the model demonstrates a distinct peripheral attention bias in these samples. The high-attention regions (highlighted in red) are predominantly concentrated along the boundaries of the tissue sections. Conversely, the core regions of the tissue—which typically harbor the dense tumor cells and critical morphological patterns required for accurate prognosis—are assigned low attention weights (indicated in blue).

**Reasoning:** This behavior suggests that in these outliers, the model may have been distracted by slide preparation artifacts (e.g., tissue folding, edge compression, or marker

residue) that frequently occur at the tissue periphery. Consequently, the model failed to capture the intrinsic tumor heterogeneity within the Region of Interest, leading to erroneous risk predictions. This finding highlights a potential direction for future improvement: incorporating boundary-aware regularization or more aggressive augmentation to suppress edge artifacts.

