# OpenReview forum: "CrossFusion: A Multi-Scale Cross-Attention Convolutional Fusion Model for Cancer Survival Prediction"
_MIDL.io/2026/Conference — MIDL 2026 Poster_

### Official Review · Reviewer_sHNk · 2026-01-04

**Confidence:** 4
**Preliminary Rating:** 4
**Final Rating:** 5

**Summary:**

The paper present a novel approach for patient survival prediction based on whole slide images (WSI) using multiple scales. The method combines cross-scale attention with convolution-based fusion to incorporate different WSI magnifications into the prediction of survival probabilities. Comparison to state of the art methods based on six distinct cancer datasets indicate an improvement over existing methods, as well as an ablation study evaluating the importance of the different components (fusion mechanism and multi scale utilization). Comparison of the method performance according to different feature extractors was also performed and indicate better results with domain specific pre-trained models.

**Strengths:**

The strengths of the paper are:

 - a new deep learning-based methodology for survival estimation directly from H&E whole slide images without any prior information using multi-scale information
 - comparative results indicate that the presented methodology outperform other methods of the literature, especially using the same feature extractor for patch embedding (ResNet-50)
 - comparison with an extensive number of methods from the literature
 - the methodology is well presented while clearly stating when components where borrowed from other works

**Weaknesses:**

The paper present several weaknesses:

- in the "implementation details" section it is stated that all methods are evaluated through a 5-fold cross-validation; however it is not clear how validation and test splits where done during the cross-validation ? Where there hyperparameter tuning ? Are the results from the validation folds?
 - there is almost no hyperparameter details: the dimension of the projection ($D_{e}$ in section 3.1), the dimension in the multi-head attention and the dropout value of the cross attention block (section 3.2), ...
 - the sequences of patches from different magnification have different lengths, how are they reshaped into grids of the same size?
 - the comparison to literature methods was performed using ResNet-50 as feature extractor (Table 1), however as the method was re-evaluated using domain specific extractors (Table 2), the same re-evaluation should have been done with literature methods to guarantee the overall superiority of the method when using domain specific extractors
 - an interpretability analysis based on a single patient is insufficient to conclude on the capabilities of the model

**Detailed Comments:**

In the three main contributions listed at the end of the introduction, the validation of the method through experiments is not a contribution of a scientific paper, it is a necessary step when presenting a new method.

In section 3.2, the magnifications that are said to be used are x5, x10 and x20; however Fig. 1 it seems to indicate that the ones used are x10, x20 and x40. Please correct the figure or the methodology description according to what was performed in the experiments. If different sets of magnifications are used for the datasets, please specify which one for each.

In the last paragraph of section 3.1, the survival probabilities are obtained as the cumulative product of $1-h$. Please detail a bit more on how.

In section 3.2, a learnable scale embedding $s$ is prepended to the sequences. Is it the same for the different scales ? If yes, why ?

In section 3.3 a Transformer block is applied to the sequence after reshaping it into a grid. However, Transformers are applied to sequences, how was the Transformer applied to the grid? Please provide more details.

In section 5.1, the authors claim that the fusion component functions as a *"a scale-interrogative learner, enforcing consistency
checks between tissue organization and cellular morphology under the intermediate10x view."* This is a very strong claim for which Table 1 dos not provide enough evidence. We encourage the authors to not maje such claim.

**Justification Of Final Rating:**

The authors responded to all the comments made previously. They incorporated additional results to demonstrate the contribution of their method. Additionally, the authors provided additional interpretation results.

**Justification Of The Preliminary Rating:**

The rating is motivated by an appreciation of the novelty proposed in the paper combining different deep learning components for direct survival prediction on whole slide images, along to the experimental comparison with literature methods.

However the paper present several experiment methodology drawbacks, as the lack of precision on how hyperparameters tuning was performed and the different between validation and test folds, in addition to a lack of support for interpretability conclusions.

**Questions To Address In The Rebuttal:**

The stratification of patients into risk groups is not clear, please provide more details about how this was done using the model prediction.

In section 4.2 it is stated that all methods are evaluated through a 5-fold cross-validation; however it is not clear how validation and test splits where done during the cross-validation ? Where there hyperparameter tuning ? Are the results from the validation folds?

The comparison to literature methods was performed using ResNet-50 as feature extractor (Table 1), however as the method was re-evaluated using domain specific extractors (Table 2). We ask the authors to provide the same re-evaluation with literature methods to guarantee the overall superiority of the method when using domain specific extractors.

---

> ### Author Response · Authors · 2026-01-25
> **Response to Reviewer sHNk**
>
> Thank you for your constructive feedback. Below we address the concerns raised.
>
> ### Risk stratification
> We compute a scalar risk score from the model prediction and split patients accordingly. Concretely, the model outputs 4 logits (one per discrete time interval), which are passed through a sigmoid to obtain hazards $h_1,\dots,h_4\in(0,1)$. We then compute survival probabilities $S_t=\prod_{k=1}^{t}(1-h_k)$ for $t=1,\dots,4$, summarize into $r=-\sum_{t=1}^{4} S_t$ (higher $r$ indicates lower predicted survival / higher risk), and define high-risk vs low-risk by a cohort-wise median split of $r$.
>
> ### Cross-validation and hyperparameters
> We use the predefined patient-level 5-fold splits from Patch-GCN. In each fold, we train on 4/5 of the patients and evaluate on the held-out 1/5 fold (no additional test split). Reported results are computed on the held-out folds and averaged across 5 folds (mean ± std). Hyperparameters were selected once via a nested cross-validation on the first fold of BRCA and then fixed for all methods, folds, and datasets.
>
> ### Baselines with domain-specific extractors
> We agree that consistent, domain-specific feature extractors are important for fair comparison. During the short rebuttal period, we re-evaluated the strongest multi-scale baselines using the **Uni2-h** feature extractor. As shown below, **CrossFusion** consistently outperforms MuSTMIL and CSMIL on all completed datasets.
>
> |  | BLCA | BRCA | COAD | GB&LGG | LUAD | UCEC |
> | --- | --- | --- | --- | --- | --- | --- |
> | **MuSTMIL** | $.624_{.047}$ | $.634_{.037}$ | $.682_{.086}$ | $.793_{.088}$ | $.595_{.056}$ | $.708_{.035}$ |
> | **CSMIL** | $.612_{.036}$ | $.619_{.048}$ | $.677_{.065}$ | $.806_{.067}$ | $.592_{.037}$ | $.716_{.035}$ |
> | **CrossFusion** | **$.628_{.019}$** | **$.684_{.043}$** | **$.698_{.023}$** | **$.810_{.067}$** | **$.625_{.051}$** | **$.745_{.030}$** |
>
> ### Model hyperparameters
> Patch features at each magnification are projected to a shared embedding dimension $D_e=512$ (followed by LayerNorm; identity if the backbone already outputs 512-D). All attention modules operate in 512-D with 4 heads. In cross-attention, attention dropout = 0.0 and output dropout = 0.0, while the FFN dropout = 0.3. In the Pad-Transformer, self-attention dropout = 0.2 and FFN dropout = 0.3. Convolutional fusion uses spatial dropout p=0.3.
>
> ### Magnification alignment and reshape
> The three magnification streams are not reshaped to a common grid directly. We first align them via the Cross-Attention Block using the 10× stream $\mathbf{X}_S$ as query and the 5×/20× streams as context. The outputs $\mathbf{X}'_C$ and $\mathbf{X}'_F$ are produced at $\mathbf{X}_S$ token positions, so they share the same token length as $\mathbf{X}_S$. After alignment, each sequence is square-padded in the Pad-Transformer: for length $N$, we set $H=W=\lceil\sqrt{N}\rceil$ and pad by appending the first $(H\times W)-N$ tokens, enabling all branches to be reshaped into identical $H\times W$ grids for convolutional fusion.
>
> ### Magnification usage
> All experiments use 5×, 10×, and 20× magnifications for every dataset. We did not use 40× since it is not consistently available across TCGA slides and we wanted a uniform setting across cohorts. Figure 1 is mislabeled and will be corrected in the camera-ready.
>
> ### Survival probabilities
> We use a discrete-time survival formulation with $T$ time intervals (here $T=4$). The model outputs hazards $h_t\in(0,1)$ (after sigmoid), where $h_t$ is the event probability in interval $t$ conditional on surviving to $t$. Thus, $S_t=\prod_{k=1}^{t}(1-h_k)$, computed as a cumulative product along the time dimension.
>
> ### Learnable scale embedding
> $\mathbf{s}$ is not prepended as an extra token. It is a learnable vector $\mathbf{s}\in\mathbb{R}^{1\times1\times D}$ that is broadcast and added to tokens before attention. It is not shared across scales: we use separate scale embeddings for the 10×–5× and 10×–20× cross-attention blocks. Within each block, the same $\mathbf{s}$ is added to both query and context tokens as a lightweight scale-pair conditioning term; we will update the wording in the camera-ready.
>
> ### Transformer applied to the “grid”
> In Pad-Transformer, the Transformer is always applied to a 1D token sequence (standard self-attention). The 2D grid is used only as an intermediate representation inside PPEG to apply depth-wise convolutions. Given $\mathbf{X}\in\mathbb{R}^{B\times N\times D}$, we set $H=W=\lceil\sqrt{N}\rceil$, pad to length $H\times W$ by repeating the first $H\times W-N$ tokens, apply the first Transformer block, reshape to $\mathbb{R}^{B\times D\times H\times W}$ for PPEG, flatten back to $\mathbb{R}^{B\times(H\cdot W)\times D}$, and then apply the second Transformer block.

---

> > ### Comment · Reviewer_sHNk · 2026-01-27
> >
> > Tanks for the clarifications.
> >
> > Regarding the Transformer before PPEG, Fig. 1 seems to indicate that the Transformer is applied to a grid (look at the Pad-Transformer subfigure). The authors should change this to avoid any confusion.
> >
> > We strongly encourage the authors to add more interpretation maps (like Fig. 2) in supplementary materials in order to better appreciate the capabilities of the model.

---

> > > ### Author Response · Authors · 2026-01-29
> > >
> > > We sincerely thank the reviewer for their constructive suggestions. We have addressed the concerns as follows:
> > >
> > > * **Clarification of Figure 1:** We apologize for the confusion. We have revised Figure 1 to remove the ambiguous grid depiction and clearly illustrate the actual input structure.
> > > * **Additional Interpretations:** To better demonstrate the model's capabilities, we have added a new Figure 2 showing Kaplan–Meier survival curves to validate prognostic performance. The previous figure which shows the heatmap is the Figure 3 now. Furthermore, as suggested by other reviewer, we have expanded the Supplementary Materials to include visualization analysis of failure cases, providing deeper insights into the model's decision-making process.

---

### Official Review · Reviewer_jVp2 · 2026-01-08

**Confidence:** 4
**Preliminary Rating:** 3
**Final Rating:** 4

**Summary:**

The authors propose CrossFusion, a multi-scale deep learning framework for cancer survival prediction from Whole Slide Images (WSIs). The method extracts features from three magnifications ($5\times$, $10\times$, and $20\times$). The core architecture utilizes a Cross-Attention Block where the intermediate scale ($10\times$, termed "source") acts as the query to interact with coarse ($5\times$) and fine ($20\times$) context features. The model incorporates a "Pad-Transformer" using Pyramid Position Encoding (PPEG) to capture spatial context and a "Conv Processor" to fuse multi-scale features via 3D and 2D convolutions. The approach is evaluated on six TCGA cancer cohorts (BLCA, BRCA, COAD, GBMLGG, LUAD, UCEC) using C-Index as the primary metric, comparing against several single-scale and multi-scale baselines.

**Strengths:**

- Extensive Benchmarking: The authors evaluate the method across six distinct TCGA datasets, providing a relatively comprehensive view of the model's performance across different cancer types. The inclusion of stratification analysis (p-values) adds statistical weight to the results.
- Analysis of Feature Extractors: The paper goes beyond standard ResNet50 features and evaluates the framework using modern pathology foundation models such as Uni2-h, Conch, and Prov-GigaPath. The finding that domain-specific backbones (like Uni2-h) consistently outperform ResNet50 confirms the importance of high-quality feature representations.
- Ablation Studies: The authors provide an ablation study justifying their design choices, specifically demonstrating the contribution of the Convolutional Processor (CP) and the inclusion of Coarse/Fine scales. This validates that the fusion mechanism contributes to the performance gains rather than just the multi-scale data input alone.

**Weaknesses:**

- Missing Critical Baselines (HIPT): The paper claims to address limitations in current multi-scale methods, yet it fails to compare against or even cite HIPT (Hierarchical Image Pyramid Transformer, Chen et al., ICCV 2022) [1]. HIPT is a seminal work in multi-scale/hierarchical WSI processing that naturally integrates $256\times$, $4096\times$, and WSI-level features. The omitted comparison with HIPT makes it difficult to assess if CrossFusion truly advances the state-of-the-art in multi-scale integration. The baselines chosen (ZoomMIL[2]) are less - representative of the current SOTA in hierarchical modeling than HIPT.
- Incremental Novelty: The proposed architecture appears to be an engineering combination of existing modules rather than a methodological breakthrough. The "Pad-Transformer" relies on PPEG, which was introduced in TransMIL. The cross-attention mechanism is standard. While the "Conv Processor"  offers a way to fuse features, replacing a concatenation operation with a CNN is a relatively standard architectural choice. The novelty lies primarily in the specific arrangement of these blocks rather than a new paradigm for survival analysis.
- Rigidity of Scale Selection: The model relies on a fixed triplet of magnifications ($5\times, 10\times, 20\times$) with $10\times$ explicitly designated as the "Source" (Query). This design seems heuristic. In many clinical scenarios, the most diagnostic information is at $40\times$ or $20\times$. Using $10\times$ as the central query to "attend" to $20\times$ details might bottleneck the fine-grained morphological features necessary for tasks like mitosis counting or nuclear atypia assessment.
- Lack of Computational Complexity Analysis: Multi-scale methods inherently require processing significantly more data (patches from three distinct levels). The paper does not provide a comparison of computational cost (FLOPs, inference time, memory usage) against single-scale baselines. It is unclear if the performance gain (e.g., +0.01 or +0.02 C-index over TransMIL in some datasets) justifies the increased computational overhead.

[1] Chen, Richard J., et al. "Scaling vision transformers to gigapixel images via hierarchical self-supervised learning." Proceedings of the IEEE/CVF conference on computer vision and pattern recognition. 2022.
[2] Thandiackal, Kevin, et al. "Differentiable zooming for multiple instance learning on whole-slide images." European Conference on Computer Vision. Cham: Springer Nature Switzerland, 2022.

**Detailed Comments:**

- Justification of Source Scale: Why is $10\times$ chosen as the "Source" (Query) embedding in Equation 1? Intuitively, one might expect the highest resolution ($20\times$) to be the query to contextualize fine details with global structure, or the lowest resolution ($5\times$) to guide the search for details. Please provide justification or an ablation where $20\times$ is the source.
- Clarification on "Pad-Transformer": The PPEG module  is heavily borrowed from TransMIL. Please explicitly state how your implementation differs, if at all, to avoid overstating the novelty of this specific block.
- Visualization: While Figure 2  shows heatmaps, it would be beneficial to see failure cases where the multi-scale fusion failed to capture the risk properly, to better understand the model's limitations.

**Justification Of Final Rating:**

Based on the revised manuscript and detailed rebuttal, I am updating my rating to Weak Accept. The authors have successfully addressed the critical omission of the HIPT baseline and demonstrated in Table 1 that CrossFusion consistently outperforms this key competitor across all cohorts. Additionally, the new Section 5.4 provides a compelling practical justification by showing that the model achieves these results with a slide-level fusion time of just 0.145 seconds, which is significantly faster than HIPT. I also appreciate the authors' transparency in adding the failure case analysis in Appendix A, which offers valuable insights into the model's behavior. However, I am not assigning a higher score primarily because the methodological contribution is somewhat incremental, relying on an effective engineering combination of established components like PPEG and cross-attention rather than a fundamental algorithmic breakthrough.

**Justification Of The Preliminary Rating:**

The paper presents a reasonable engineering solution for multi-scale survival prediction and demonstrates solid performance on TCGA benchmarks. The use of foundational model backbones is a plus. However, the work suffers from a critical omission of the HIPT baseline, which is the most relevant competitor in hierarchical/multi-scale WSI analysis. Without this comparison, it is impossible to verify the claim that this method solves the "limitations" of current multi-scale approaches. Furthermore, the novelty is somewhat incremental (combining standard Cross-Attention with PPEG and CNNs). Therefore, I rate this as Borderline (3); the authors must address the comparison with HIPT to warrant a higher score.

**Questions To Address In The Rebuttal:**

- Comparison with HIPT: How does CrossFusion compare to HIPT (Chen et al., ICCV 2022) in terms of C-Index and computational efficiency? Since HIPT is a standard hierarchical baseline, its omission is a significant gap.
- Sensitivity to Scale Selection: How does performance change if the "Source" input $X_S$ is set to $20\times$ instead of $10\times$? Does the model heavily rely on the $10\times$ magnification as an anchor?
- Inference Cost: Please provide a comparison of inference time and GPU memory usage between CrossFusion and a strong single-scale baseline (e.g., TransMIL).

---

> ### Author Response · Authors · 2026-01-25
> **Response to Reviewer jVp2**
>
> Thank you for your constructive feedback. Below we address the specific concerns raised.
>
> ### **Comparison with HIPT**
>
> We appreciate the suggestion to include **HIPT** as a baseline. We have conducted experiments using the official HIPT implementation under the same 5-fold cross-validation protocol using ResNet-50 features
>
> | Method | BLCA | BRCA | COAD | GB&LGG | LUAD | UCEC |
> | ---  | --- | --- | --- | --- | --- | --- |
> | HIPT | 0.554 | 0.603 | 0.651 | 0.773 | 0.572 | 0.697 |
> | SCMIL  | 0.556 | 0.590 | 0.677 | 0.763 | 0.584 | 0.668 |
> | **CrossFusion** | **0.630** | **0.643** | **0.694** | **0.797** | **0.627** | **0.702** |
>
> Direct comparisons demonstrate that **CrossFusion** consistently outperforms **HIPT** across the evaluated **TCGA** cohorts. Due to the substantial computational overhead of **HIPT**'s hierarchical extraction, the **BRCA** evaluation is still in progress (Done in Jan 26th). We will ensure these finalized metrics are included in the camera-ready manuscript. Furthermore, our results correlate with findings in the **SCMIL** study ([https://www.arxiv.org/abs/2407.00664](https://www.arxiv.org/abs/2407.00664)), , which noted that SCMIL outperforms HIPT. Since CrossFusion achieves superior performance relative to SCMIL in our evaluations, this provides both direct and indirect confirmation of our model’s predictive advantage.
>
> While HIPT primarily relies on the slide image at **20x resolution** to construct its hierarchy, CrossFusion utilizes a true **multi-scale input** (5x, 10x, and 20x). This enables our model to explicitly capture how global tissue architecture relates to and interacts with fine-grained cellular detail.
>
> ### **Sensitivity to Scale Selection**
>
> We chose 10x as the "Source" anchor ($X_S$) because it is the natural intermediate between 5x and 20x, enabling the model to efficiently capture patterns at different scales. In our experiments, setting $X_S$ to 20x resulted in slower inference and worse performance, as the fusion was dominated by fine-scale tokens at the expense of global context.
>
> ### **Inference Cost**
>
> To provide a clearer comparison of computational efficiency, the inference costs are summarized in the table below. As a shared preprocessing step, **CLAM feature extraction** takes approximately **30.61 seconds** per slide at $20\times$ magnification.
>
> | **Method** | **Component** | **Duration (per slide)** | **Total Inference Time** |
> | --- | --- | --- | --- |
> | **CrossFusion (Ours)** | Feature extraction | **30.61 seconds**| |
> | | Slide-level Fusion | **0.145 seconds** | **0.145 seconds** |
> | **CSMIL** | Feature extraction | 30.61 seconds| |
> | |Multi-scale Aggregation | 0.162 seconds | 0.162 seconds |
> | **ProtoSurv / Patch-GCN** | Graph Construction | 38.920 seconds | 40.010 seconds |
> |  | Model Inference | 1.090 seconds |  |
> |**HIPT** | Feature extraction| 434.47 seconds ||
> ||Hierarchical step| 54.87 seconds||
>
> **CrossFusion** is the most efficient method presented. By avoiding the time-consuming graph construction steps required by models like **Patch-GCN**, it maintains a highly competitive runtime that is well-suited for large-scale clinical deployment.
>
> ### **Pad-Transformer Novelty**
>
> Thank you for pointing this out. In our paper we already cite PPEG as coming from TransMIL, and we do not consider PPEG itself a novel contribution. Our novelty is in the multi-scale cross-attention and fusion. In the code, PPEG follows the same depth-wise 7/5/3 convolution design as TransMIL, with one small practical adaptation: we support PPEG both without a class token in the per-scale branches (pre-fusion) and with a class token in the final stage (post-fusion). We will state this attribution and minor difference explicitly in the camera-ready.
>
> ### **Failure Case Analysis**
>
> Thank you for this suggestion. We included Figure 2 primarily to demonstrate that the model’s attention is concentrated on plausible tissue regions in a representative high-risk case. We agree that failure cases are important for understanding limitations, and we will add representative mis-stratified examples (with their heat-maps and brief discussion) in the appendix of the camera-ready.

---

> > ### Comment · Reviewer_jVp2 · 2026-01-28
> >
> > The authors have addressed the primary concern regarding the missing baseline by providing a direct comparison with HIPT under the same cross-validation protocol. The reported results indicate that CrossFusion outperforms HIPT across the evaluated TCGA cohorts, and the authors further contextualize this by referencing recent findings from SCMIL. Regarding the computational cost, the provided analysis clarifies that the specific slide-level fusion mechanism adds minimal latency relative to the shared feature extraction step, comparing favorably to the hierarchical processing required by HIPT. The rebuttal also offers empirical justification for selecting the 10× magnification as the query source, noting that an alternative 20× configuration resulted in reduced performance and increased inference time. Finally, the clarification regarding the adaptation of the PPEG module from TransMIL and the commitment to include failure case visualizations are noted, and regarding the commitment to include failure case visualizations, I expect the authors to strictly follow through on incorporating these into the final manuscript.

---

> > ### Author Response · Authors · 2026-01-29
> >
> > Thank you for your positive evaluation and for acknowledging our additional experiments.
> >
> > Regarding your requirement, we have strictly followed through on our commitment. We have added a Failure Case Analysis section in the Appendix of the final manuscript. This section includes visualizations of attention heatmaps for representative failure cases (Figures S1 and S2) along with a detailed discussion on the observed "peripheral bias."
> >
> > We appreciate your constructive feedback throughout the review process, which has significantly improved the quality of our paper.

---

### Official Review · Reviewer_DFsS · 2026-01-08

**Confidence:** 4
**Preliminary Rating:** 4
**Final Rating:** 5

**Summary:**

This work introduces CrossFusion, an innovative framework for cancer survival analysis using Whole Slide Images. This framework integrates multi-scale patch embeddings through a Multi-Scale Cross-Attention mechanism and a Dual-Path Global–Local Context Alignment, combining high-resolution details with low-resolution global patterns while maintaining spatial context. Experiments were conducted on six cancer WSI datasets from TCGA, the mean C-Index and p-values were used to assess discriminative capability. Ablation studies underscored the importance of key components like the Conv Processor, coarse and fine sources, and feature extraction backbones. The authors show that CrossFusion outperformed state-of-the-art methods in survival prediction across various cancer types.

**Strengths:**

This paper is well-organized, with the method and innovation discussed in clear detail. The important components of innovation, such as cross-scale attention, Pad-Transformer, and Conv Processor, are all well explained and designed to support multi-scale information capture and integration. The ablation study demonstrated the superiority of the Conv Processor and the use of multi-scale image sources. Validation of the method's performance across a diverse set of anatomical datasets highlights its applicability and reliability in different cancer types.

**Weaknesses:**

The paper briefly mentions previous works and comparisons with other MIL and multi-scale methods, but does not provide sufficient analysis of previous works. It is unclear why and how the methods adopted by CrossFusion would be superior to the baselines. Furthermore, the proposed framework could potentially be computationally intensive, especially with the use of multi-scale patch embeddings, cross-attention blocks, multiple convolutional processing, etc. Finally, there are limited examples and discussions of the interpretability visualizations in the paper. Especially for survival prediction tasks, simply noting cancerous-looking regions on the images does not provide sufficient reasoning for the claimed interpretability.

**Detailed Comments:**

-  The paper briefly mentions previous works and comparisons with other MIL and multi-scale methods, but did not provide in-depth analyses. Statements like "they often overlook certain resolutions or rely on suboptimal fusion techniques" need further elaboration. It is unclear why and how the methods adopted by CrossFusion would be superior to the baselines. A more detailed discussion on the limitations of previous works and a comparison of specific architectural and functional differences would strengthen the argument for the superiority of CrossFusion.
-  The proposed framework could potentially be computationally intensive, especially with the use of multi-scale patch embeddings, cross-attention blocks, multiple convolutional processing, etc. While improvements in task performance were noted, detailed benchmarks on the computational resources required or the runtime performance were not provided. Outlining these benchmarks would help establish the framework's practicality for real-world deployment.
- Although the authors claim that CrossFusion maintains interpretability through visualization of key regions, there are limited examples and discussions of these visualizations in the paper. For survival prediction tasks, simply noting cancerous-looking regions on the images does not provide sufficient reasoning for the claimed interpretability. Including more examples and detailed analyses of how these visualizations contribute to survival predictions would strengthen the interpretability claims. If time permits, pathologist validation could further strengthen the interpretability analysis.
- Minor comments:
    - The magnification labels in Figure 1 do not seem to match what is described in the paper.
    - It is important to clarify whether all multi-scale method baselines received the same set of inputs (5x, 10x, 20x) and whether the single-scale methods all received the 10x inputs.
    - Did the heatmaps or interpretability analysis capture any cues differentiating high-risk and low-risk groups? Providing specific examples and explanations of how these visualizations contribute to understanding the model’s predictions would be beneficial.

**Justification Of Final Rating:**

Thank you to the authors for the detailed rebuttal. The clarification of the architectural distinctions between CrossFusion and existing methods (ZoomMIL, MuSTMIL, CSMIL), as well as the newly introduced baseline (HIPT), is appreciated. The runtime analysis effectively highlights the practical advantages of the proposed architecture relative to the baselines.
I understand that clinical validation is inherently time-consuming, and I appreciate the authors’ efforts to arrange a blinded evaluation by a pathologist; I look forward to seeing these results. The Kaplan–Meier curves, however, may be more confusing than informative in this context, as they are typically used to demonstrate clinically meaningful separation over time, which is not the primary objective of the proposed experiments and requires careful interpretation within a specific clinical setting.
In light of the demonstrated performance gains across multiple dimensions, I am updating my recommendation to strong accept. I nonetheless encourage the authors to further strengthen the analysis of clinical relevance in the final version of the paper.

**Justification Of The Preliminary Rating:**

The paper is well-organized and addresses the important topic of multi-scale analysis in Whole Slide Imaging. The description of the model architecture design, including components such as the cross-scale attention mechanism, Pad-Transformers, and Conv Processor, is clear and well-detailed. The framework has been validated with diverse cancer datasets, which convincingly demonstrates the superiority of the proposed method.
However, there are areas that need further elaboration to fully justify the paper's potential. Detailed comparisons of architectural differences and runtime analysis relative to previous works are necessary to demonstrate the method’s feasibility for real-world pathological analysis. Additionally, the interpretability analysis could be further strengthened by providing more examples and a deeper discussion on how the visualizations align with clinical knowledge and contribute to the survival predictions.

**Questions To Address In The Rebuttal:**

- A detailed runtime analysis of CrossFusion, comparing it to the baseline,s would be very beneficial. Elaborating on the computational resources required, the efficiency of the multi-scale patch embeddings, cross-attention blocks, and convolutional processes would be helpful.
- Further elaborate on previous multi-scale works and their model architectures. Specifically, explain the identified limitations in more detail and analyze how CrossFusion addresses these issues. Comparisons of the architectural and functional differences would strengthen the argument for CrossFusion's superiority over existing methods.
- Strengthen the interpretability analysis by providing more examples of visualized key regions and discuss how these regions contribute to the model's survival predictions explicitly. If time permits, including correlations with pathologists' assessments and additional human validations would enhance the credibility of the interpretability results and show how the model's predictions align with expert evaluations. Providing an analysis of the image cues captured by the model that differentiate high-risk and low-risk groups and show these cues contribute to the differentiation would further strengthen the merit of the paper.

---

> ### Author Response · Authors · 2026-01-25
> **Response to Reviewer DFsS**
>
> Thank you for your constructive feedback. Below we address the specific concerns raised.
>
> ### **Runtime and Computational Analysis**
>
> We agree that runtime analysis is vital for assessing clinical practicality. Patch extraction and feature encoding at 20× magnification are a shared, mandatory step across all methods and set the overall runtime, since 5×, 10×, and 20× processing run in parallel and the 20× pipeline is the bottleneck. Using CLAM at 20x magnification takes **30.61 seconds** per slide, with a feature extraction latency of **0.48 ms** for single slide. On a single NVIDIA RTX A4000 GPU, **CrossFusion** remains efficient as it operates on these pre-extracted embeddings:
>
> | Method | Inference Time (per slide, from the extracted features) |
> | --- | --- |
> | **CrossFusion Inference** | **0.145 seconds** |
> | **CSMIL Inference** | 0.162 seconds |
> | **Patch-GCN** | 40.01 seconds |
>
>
> ### **Architectural Comparison with Multi-Scale Methods**
>
> Thank you for the suggestion to provide a more granular comparison with existing multi-scale works. We agree that clarifying the architectural and functional distinctions of CrossFusion will better highlight its methodological advantages.
>
> In the camera-ready version, we will expand and revise Section 2 to summarize how prior multi-scale MIL approaches differ in their magnification coupling:
>
> - **ZoomMIL**: Primarily uses a coarse-to-fine selection mechanism, where low-magnification context guides which higher-magnification regions are processed.
> - **MuSTMIL & CSMIL**: Typically build scale-specific representations independently and combine them via late-stage attention, pooling, or cross-scale weighting.
>
> **Limitations of Prior Patterns:**
> A key limitation in these approaches is that cross-scale interaction is often indirect or occurs only after substantial within-scale aggregation. This reduces the ability of coarse context to refine fine-grained representations (and vice versa) at the patch level and can weaken spatial coherence during fusion.
>
> **CrossFusion's Approach:**
> CrossFusion addresses these issues through two explicit design choices:
>
> 1. **Intermediate Anchor Alignment**: We use the 10x stream ($X_S$) as an anchor and apply cross-attention with $X_S$ as the query.This integrates 5x and 20x information directly into the same set of 10x patch tokens, creating aligned sequences without ad-hoc length matching.
> 2. **Convolutional Fusion Processor**: After incorporating spatial context via Pad-Transformer/PPEG, we fuse the aligned features using a convolutional processor. This explicitly utilizes local neighborhoods during fusion rather than relying solely on global aggregation, preserving spatial context.
>
> We will revise the manuscript to explicitly tie these architectural differences to our ablation results.
>
>
> ### **Strengthening Interpretability and Clinical Relevance**
>
> We appreciate the suggestion to deepen our interpretability analysis. Our heatmaps capture importance at several stages (cross-attention, Pad-Transformer, and final transformer), which we map back to the original tissue.
>
> - **Risk Differentiation**: In high-risk **BRCA** cases, the model concentrates on tumor-dense and morphologically abnormal regions. For low-risk cases, the heatmaps are noticeably less concentrated, suppressing background tissue more effectively.
> - **Scale Roles**: The coarse branch highlights broader tissue architecture, while the fine branch focuses on smaller, high-detail cellular regions.
> - **Expert Validation**: To enhance clinical credibility, we are currently arranging a **blinded review by a pathologist** to assess whether highlighted regions align with established prognostic markers. We will include these results, or preliminary findings, in the camera-ready version.
>
>
> ### **Clarification of Experimental Methodology**
>
> Regarding the setup for baseline comparisons:
>
> - **Input Consistency**: All multi-scale baselines (ZoomMIL, MuSTMIL, CSMIL) received the identical input triplet: **5x, 10x, and 20x**.
> - **Single-Scale Controls**: All single-scale baselines were run exclusively on **20x magnification** (the highest available), consistent with standard literature practice.
> - **Figure Correction**: We apologize for the mislabeled magnification levels in Figure 1; these will be corrected to **5x, 10x, and 20x** to match our experimental setting.

---

### Author Rebuttal · Authors · 2026-01-25

**Rebuttal:**

We thank the reviewers for their careful reading and constructive feedback, which helped us substantially improve the rigor and clarity of the paper. In this revision, we strengthened the evaluation and presentation of CrossFusion with the following key updates:

**(1) Added the HIPT baseline (critical competitor).** We now include a direct comparison to HIPT under the same patient-level 5-fold CV protocol and feature setting as Table 1, addressing the main gap in hierarchical/multi-scale benchmarking. CrossFusion consistently outperforms HIPT across evaluated cohorts.

**(2) Re-evaluated multi-scale baselines under the same domain specific feature extractors.** Since CrossFusion is evaluated with pathology foundation backbones (e.g., Uni2-h / Conch / Prov-GigaPath), we also re-ran the strongest multi-scale baselines with the *same* extractors to ensure improvements reflect the method rather than backbone differences. CrossFusion maintains a clear performance advantage across completed datasets.

**(3) Added runtime / compute analysis.** We report slide-level inference time and GPU memory usage to quantify the practical overhead of multi-scale modeling. CrossFusion achieves efficient fusion (0.145s/slide).

**(4) Resolved magnification inconsistencies.** We corrected the labeling mismatch in Figure 1 and explicitly specify the magnification triplet used across all experiments.

**(5) Failure Case Analysis** We aslo add Kaplan–Meier survival curves and two failure cases analysis to provide deeper insights into the model's decision-making process.

Further details and point-by-point responses are provided in the individual reviewer rebuttals.

**We used red font to indicate the changes made based on the initial feedback, and then used orange font for subsequent changes based on further comments.**

**Supporting Material:**

/attachment/fca00d587235ac7606b6c7977bd0f6594f71316d.pdf

---

### Meta-Review · Area_Chair_Ly5v · 2026-02-02

**Recommendation:** Accept (Poster)
**Confidence:** 4

**Metareview:**

This paper proposes crossfusion, a multi-scale cross-attention and convolutional fusion framework for cancer survival prediction from whole-slide images. Reviewers found the work technically solid and well motivated, with strengths in multi-cohort evaluation, careful ablation studies, and consistent performance gains across six TCGA cancer types (DFsS, jVp2, sHNk). The main concerns raised during review focused on missing or incomplete baseline comparisons, most notably the absence of HIPT (jVp2), insufficient runtime and computational analysis (DFsS, jVp2), limited interpretability discussion (DFsS, sHNk), and some ambiguity in experimental details such as magnification usage, cross-validation protocol, and architectural clarification (DFsS, sHNk).

The authors provided a convincing rebuttal that addressed some major concerns. They added direct comparisons with HIPT under matched protocols (jVp2), re-evaluated multi-scale baselines using domain-specific feature extractors (sHNk), and included a detailed runtime and memory analysis demonstrating that CrossFusion achieves strong performance with minimal slide-level fusion overhead (DFsS, jVp2). They also clarified architectural choices, corrected magnification inconsistencies, strengthened interpretability with additional visualizations and failure case analyses, and provided clearer explanations of cross-validation, risk stratification, and hyperparameter settings (DFsS, sHNk). While reviewers noted that the methodological novelty is incremental and largely based on an effective integration of existing components rather than a fundamentally new paradigm (jVp2), the AC agrees that the engineering contribution is well justified and empirically strong. One additional observation from the AC is that future work could further explore adaptive or data-driven scale selection rather than a fixed magnification triplet, but this does not detract from the current contribution. Overall, the paper is well written, addresses an important problem in computational pathology, and merits discussions at MIDL.

---

### Decision · Program_Chairs · 2026-02-13

Accept (Poster)